# Fault-Related Fluid Flow Implications for Unconventional Hydrocarbon Development, Beetaloo Sub-Basin (Northern Territory, Australia)

Emanuelle Frery [1,*], Conor Byrne [2], Russell Crosbie [3], Alec Deslandes [3], Tim Evans [4], Christoph Gerber [3], Cameron Huddlestone-Holmes [1], Jelena Markov [1], Jorge Martinez [3], Matthias Raiber [3], Chris Turnadge [3], Axel Suckow [3] and Cornelia Wilske [3]

1   CSIRO Energy, 26, Dick Perry Avenue, Kensington, WA 6152, Australia; cameron.hh@csiro.au (C.H.-H.); jelena.markov@csiro.au (J.M.)
2   Byrne Geophysique, Perth, WA 6018, Australia; byrneconor@mac.com
3   CSIRO Land & Water, Waite Road, Urrbrae, SA 5064, Australia; russell.crosbie@csiro.au (R.C.); alec.deslandes@csiro.au (A.D.); christoph.gerber@csiro.au (C.G.); jorge.martinez@csiro.au (J.M.); matthias.raiber@csiro.au (M.R.); chris.turnadge@csiro.au (C.T.); axel.suckow@csiro.au (A.S.); corneliamaria.wilske@csiro.au (C.W.)
4   Geoscience Australia, 101 Jerrabomberra Avenue, Symonston, Canberra, ACT 2609, Australia; timthegeo@internode.on.net
*   Correspondence: emanuelle.frery@csiro.au

**Abstract:** This study assesses potential geological connections between the unconventional petroleum plays in the Beetaloo Sub-basin, regional aquifers in overlying basins, and the near surface water assets in the Beetaloo Sub-basin Northern Territory, Australia. To do so, we built an innovative multi-disciplinary toolbox including multi-physics and multi-depth imaging of the geological formations, as well as the study of potentially active tectonic surface features, which we combined with measurement of the helium content in water sampled in the aquifer systems and a comparative analysis of the surface drainage network and fault lineaments orientation. Structures, as well as potential natural active and paleo-fluid or gas leakage pathways, were imaged with a reprocessing and interpretation of existing and newly acquired Beetaloo seismic reflection 2D profiles and magnetic datasets to determine potential connections and paleo-leakages. North to north-northwest trending strike slip faults, which have been reactivated in recent geological history, are controlling the deposition at the edges of the Beetaloo Sub-basin. There are two spring complexes associated with this system, the Hot Spring Valley at the northern edge of the eastern Beetaloo Sub-basin and the Mataranka Springs 10 km north of the western sub-basin. Significant rectangular stream diversions in the Hot Spring Valley also indicates current or recently active tectonics. This suggests that those deep-rooted fault systems are likely to locally connect the shallow unconfined aquifer with a deeper gas or fluid source component, possibly without connection with the Beetaloo unconventional prospective plays. However, the origin and flux of this deeper source is unknown and needs to be further investigated to assess if deep circulation is happening through the identified stratigraphic connections. Few north-west trending post-Cambrian fault segments have been interpreted in prospective zones for dry gas plays of the Velkerri Formation. The segments located in the northern part of the eastern Beetaloo Sub-basin do not show any evidence of modern leakages. The segments located around Elliot, in the south of the eastern Beetaloo Sub-basin, as well as low-quality seismic imaging of potential faults in the central part of the western sub-basin, could have been recently reactivated. They could act as open pathways of fluid and gas leakage, sourced from the unconventional plays, deeper formations of the Beetaloo Sub-basin or even much deeper origin, excluding the mantle on the basis of low $^3$He/$^4$He ratios. In those areas, the data are sparse and of poor quality; further field work is necessary to assess whether such pathways are currently active.

**Keywords:** structural geology; fluid flow along faults; seismic; helium isotopic measurements

## 1. Introduction

Faults are natural pathways for fluids within the crust [1,2] opened by earthquakes, overpressure or pressure-dissolution [3], and progressively closed by mechanical or chemical processes [4–7].

The impact of human activities on the creation or reactivation of fault pathways need to be assessed to provide high levels of confidence to the community that unconventional hydrocarbon resource exploration can be safely carried out with very low risk of environmental issues, such as groundwater contamination, groundwater drawdown, subsidence, or induced seismicity [8–12].

Felt seismic events induced by hydraulic stimulation have been documented [13–15] with few occurrences linked to unconventional resources hydraulic stimulation [16–19]. Those rare occurrences are linked to local conditions, such as fluid-pressure changes in critically stressed faults [20]. Geomechanical experiments reveal that hydraulic stimulation can reactivate pre-existing faults that are weak and optimally oriented to the stress field [21,22].

Most unconventional hydrocarbon resources are located onshore with poor coverage by seismic datasets, being mostly 2D seismic reflection profiles focused on the prospective plays, partially imaging the fault pathways between the deep resource plays and the shallower aquifers. Pre-Cambrian basins typically experienced several episodes of major deformations and exhumations (27) and it is challenging to define the fault's architecture, internal properties, and role of natural fluid circulation at the basin scale.

This study assesses the level of knowledge of natural potential connections between the unconventional petroleum plays, the overlying aquifers, and the surface within the Beetaloo region and discuss the potential of those pathways to be reactivated or altered by the unconventional petroleum exploration and production activity.

This study presents an innovative multi-disciplinary toolbox, including multi-physics and multi-depth imaging of the geological formations, as well as potentially active tectonic surface features and measurement of the helium content in water sampled in the aquifer systems. We imaged the structures as well as potential natural active and paleo-fluid or gas leakage pathways with a new processing and interpretation of the existing and newly acquired Beetaloo seismic reflection 2D profiles magnetic datasets.

## 2. Beetaloo Region (Beetaloo Sub-Basin): Prospectivity, Regional Structural Framework, and Groundwater Systems

The Beetaloo region uses the geological boundaries of the Beetaloo Sub-basin to delineate a region covering all the geological units located above the geophysical basin [12]. The Beetaloo Sub-basin is a structural component of the greater McArthur Basin in Northern Territory (NT), Australia (Figure 1A), and contains more than 5000 m of Mesoproterozoic Roper Group sedimentary rocks which are currently being explored for shale gas, tight gas, and shale oil resources [23–26] highlighted that—subject to further exploration, resource assessment, and infrastructure development—unconventional resources production is feasible in the Beetaloo Sub-basin within 5 to 10 years. This covers the shale gas plays of the Kyalla Formation and the Amungee Member of the Velkerri Formation, along with a tight sandstone play in the Hayfield sandstone member of the Hayfield Mudstone [12,26,27]. Production tests already report 6.6 Tcf of contingent gas resource at the Amungee NW-1H exploration well in the organic-rich shales of the Velkerri Formation and an average rate of 250,000 cfpd over a 17-day test period from the Carpentaria-1 exploration well. A third well, Tanumbrini 2H, has also recently begun to be spudded [28–32].

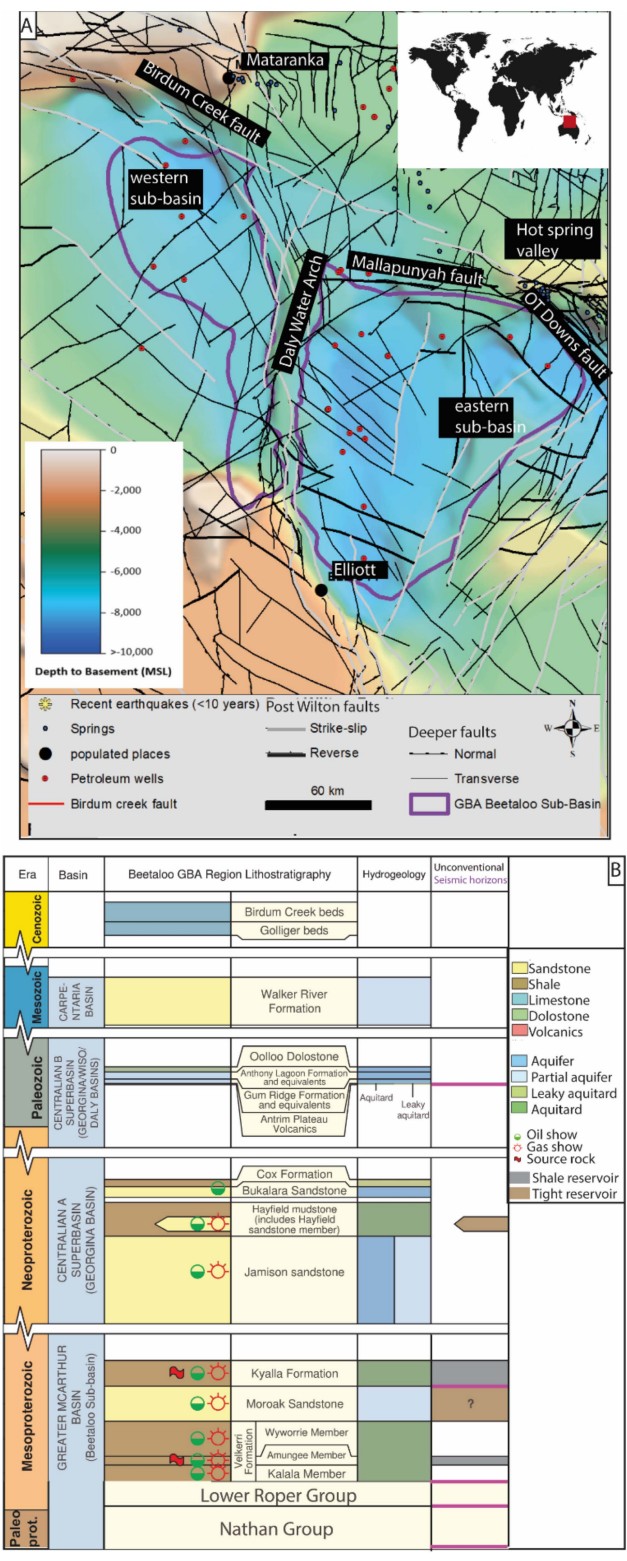

**Figure 1.** (**A**) Location and structural context of the Beetaloo Sub-basin. Basemap at the base of the Roper Group and basement (SEEBASE®) from Frogtech Geoscience (2018b) and Frogtech Geoscience (2018a) showing the Beetaloo-Basin sedimentary depocenters at a regional scale with major fault trends, recorded earthquakes, and active springs [27,33,34] (**B**) Hydrostratigraphic column for the Beetaloo GBA region. The chart breaks at 1300 to 840 Ma and 480 to 120 Ma are for display purposes [12].

The Beetaloo Sub-basin is bounded by several prominent structural highs associated with faults and springs (Figure 1A). The sub-basin is divided into eastern and western depocenters by the Daly Waters Arch and is completely buried by overlying sedimentary basins, including the Neoproterozoic to Cambrian Georgina, Wiso and Daly basins, and the Mesozoic Carpentaria Basin (Figure 1B). These younger sedimentary basins are in places overlain by Cenozoic sediments [27].

The sedimentary units that comprise the Beetaloo Sub-basin are affected by three main faulting events which correspond to each of the Paleoproterozoic and Mesoproterozoic eras [35–37]. These faults were reactivated at different stages of geological history and inverted during the Early Mesoproterozoic Syn-Isan Orogeny, the Post Roper inversion, and the Peterman Orogeny [38]. The NT Geological Survey produced a structural interpretation of the greater McArthur Basin [39] using the Proterozoic depositional packages as the stratigraphic framework (Northern Territory Government, 2019) with the most recent fault package interpreted from the geophysical dataset being post-Wilton faults (Figure 1A), fault linked with a Middle to late Mesoproterozoic non-deposition, and inversion event linked with a Mesoproterozoic orogenesis that affected large tracts of central Australia between 1300 and 1050 Ma [27,39].

The hydrogeology of the Beetaloo Geological and Bioregional Assessment (GBA) region was conceptualised as three discrete groundwater subsystems (Figure 1B): (i) confined and partially confined aquifers in Neoproterozoic rocks and the overlying Cambrian volcanics of the Kalkarindji Suite; (ii) the Cambrian Limestone Aquifer (CLA) system that is the most significant aquifer system in the region for the pastoral industry, community, and environment; and (iii) where saturated, the Cretaceous Carpentaria Basin and Cenozoic sediments. These aquifer systems are underlain by the sedimentary rocks of the Roper Group, hosting the key unconventional hydrocarbon resources, which, for the most part, are inferred to be aquitards [12,27,40].

## 3. Methods

### 3.1. Geophysical Imaging: Magnetic Signal Transformations

The Beetaloo Sub-basin has been covered by a number of aeromagnetic surveys, flown by the NTGS, Geoscience Australia, and the private sector since the mid 1960s. All of these data are compiled and gridded at 80 m cell width. The data are used to create a magnetic map of the Northern Territory (NT) which is updated every few years with the newly available information. The latest version of the map was made available in 2020 [41]. The texture and amplitude variation of the magnetic signal strongly depend on various structural (faults, folds) and lithological factors (such as the variable distribution of the magnetic minerals among different units). To better understand the causative sources, a few transformation techniques can be applied to the aeromagnetic data (Table 1). The short wavelength features produced by shallow sources are enhanced using the first vertical derivative transformation (1VD); long wavelength sources generally caused by deeper bodies are highlighted by a low pass filter (LPF). Prior to application of these enhancements, reduction to pole transformation (RTP) is applied to the aeromagnetic data. All these transformations are further described in Table 1. The processing of the aeromagnetic data were undertaken in Geosoft Oasis Montaj.

**Table 1.** Magnetic signal transformation types.

| Transformation | Description | Purpose | Disadvantage | Reference |
|---|---|---|---|---|
| Reduction to pole (RTP) | Removes asymmetrical influence of the ambient magnetic field and reproduces magnetic anomalies assuming vertical field | In the vertical ambient field magnetic anomalies are easier to understand. Magnetic bodies with positive susceptibility contrast appear under the positive magnetic anomalies | Transformation fails if there is presence of remanent magnetisation which is not accounted for | [42,43] |
| First Vertical Derivative (1VD) | Produces maximum rate of change of the magnetic field in the vertical direction | Used to emphasise high frequency, usually shallow magnetic sources | Can highlight noise or survey artefacts | [44] |
| Low Pass Filter (LPF) | Reduces contribution from the high frequency part of the spectrum | Used to emphasise low frequency magnetic features | Can introduce edge effects and other processing artefacts to the data | [44] |
| Analytic Signal Amplitude (ASA) | Produces function maxima over magnetic bodies dependent on the magnetisation intensity | Highlights magnetic anomalies of the highest magnetisation independent of the magnetisation direction | If the remanent component of the field is large, the ASA can be affected | [42,45–47] |

### 3.2. Geophysical Imaging: Seismic Reflection 2D Profiles

For the present study, 189 seismic reflection 2D profiles totaling over 8500 km of 2D seismic lines publicly available in the Beetaloo Sub-basin [48] were loaded into a Petrel® project with a Seismic Reference Datum of 0 m and a Replacement Velocity of 2000 m/s.

The vintages of seismic data range from 1989 until the latest survey in 2015 and have a varying quality from very poor to excellent. The seismic reference datum and the replacement velocity used to process the seismic lines were checked for consistency. In many cases, the seismic datum is known but the replacement velocity is unknown, or not reported in the SEGY header, which results in uncertainty in the time shifts of the seismic data affecting misties. This was overcome by trying to correct and minimise the misties on a line-by-line basis as well as correcting the mistie while interpreting.

All publicly available geological and geophysical data over the Beetaloo region were compiled [27] and used to interpret the shallow horizons and faults. This interpretation focuses on the shallow regions, as well as some deeper horizons, from above the Velkerri Formation to define the fault network for the shallow section of the Beetaloo region. Horizons and faults were interpreted, including with fault polygons for five horizons, creating TWT structure surfaces which were depth converted by the Petrel convergent method with a 200 m xy step and calibrated with 26 petroleum wells.

### 3.3. Helium Measurements

Groundwater samples for the measurement of helium were obtained with an uncertainty of 1-sigma from stock water supply and monitoring bores as part of environmental tracer studies in the area [49,50]. A Bore Boss pump (Model no: BBR300S) or a Grundfos MP1, depending on bore diameter, was used for bores without any pump headworks. For bores with headworks, water samples were collected from existing sampling taps or a fitted outlet on the pump headworks. Bores were purged for a minimum of three to five bore volumes. Where this method was not practical due to the size of the flow or headworks configuration, bore yield was obtained from information provided in the NT groundwater database (Northern Territory Government, 2019). During purging, field parameters were monitored regularly, and groundwater samples were collected only when the field parameters (temperature, electrical conductivity (EC), dissolved oxygen, and pH) were

stabilised within a range outlined in the NT Government methodology for the sampling of groundwater advisory note [51]. Samples for noble gases were collected using copper tubes and clamps developed by CSIRO, similar to established cold-seal methodologies [52].

Noble gas samples were analysed at the CSIRO Environmental Tracer Laboratory (ETL). Dissolved gasses were first separated from water on an offline extraction system. The resulting gas subsamples were analysed using a fully automated noble gas facility. The analysis includes drying of all gases, raw gas analysis on a quadrupole mass spectrometer, separation of noble gases from reactive gases using a variety of reactive getter systems, separation of the noble gases by cryo techniques, and measurement of gas amounts and their isotopic composition using a spinning rotor gauge, quadrupole mass spectrometers, and a high-resolution HelixMC Plus (Thermo) noble gas mass spectrometer [53]. Post processing of all noble gas measurements was carried out using the LabData laboratory information management and database system [54]. To ensure reproducibility of the results, duplicate, and in some cases triplicate samples were collected and analysed.

### 3.4. Fault and Surface Drainage Network Delineation (Tectonic Geomorphology)

It is well accepted that the course of rivers often follows zones of structural weakness at the Earth's surface, and that geological structures and slope are primary controls of the spatial arrangement of drainage channels [55,56]. Consequently, in regions with limited surface outcrops, such as the Beetaloo Sub-basin, river channel patterns represent a very useful guide to review and identify underlying geological structures. This has led to the recognition that geotectonics activity occurs in many sedimentary basins throughout Australia and elsewhere, and that many of the major deep basement structures have been reactivated during recent geological times [40,57–59]. In sedimentary basins, geotectonics activity can often be observed by the presence of catchment constrictions [40,59] or rectangular (or more generally high angle) diversions of the course of streams.

The analysis of possible relationships between geomorphological features represented by drainage patterns with regional structural trends observed in mapped geological faults from different sources was carried out in this study to provide additional evidence in examining the role of faults as potential connectivity pathways.

The drainage network dataset used was automatically extracted from a 1 s digital (~30 m) elevation model (SRTM, Geoscience Australia and CSIRO Land and Water, 2010) using the Watershed Generation workflow within the Global Mapper (version 21) platform (Figure 2).

Fault line datasets [39,60,61] were interpreted separately and compared against the drainage patterns from the dataset described above.

Rose diagrams, which show the orientation of faults and surface drainage features, were then created using Rockworks software (version 17) as a graphical analytical tool to visually compare and analyse spatial trends in surface water drainage and fault networks. The results of this analysis were then further compared with other lines of evidence, such as the location of mapped springs.

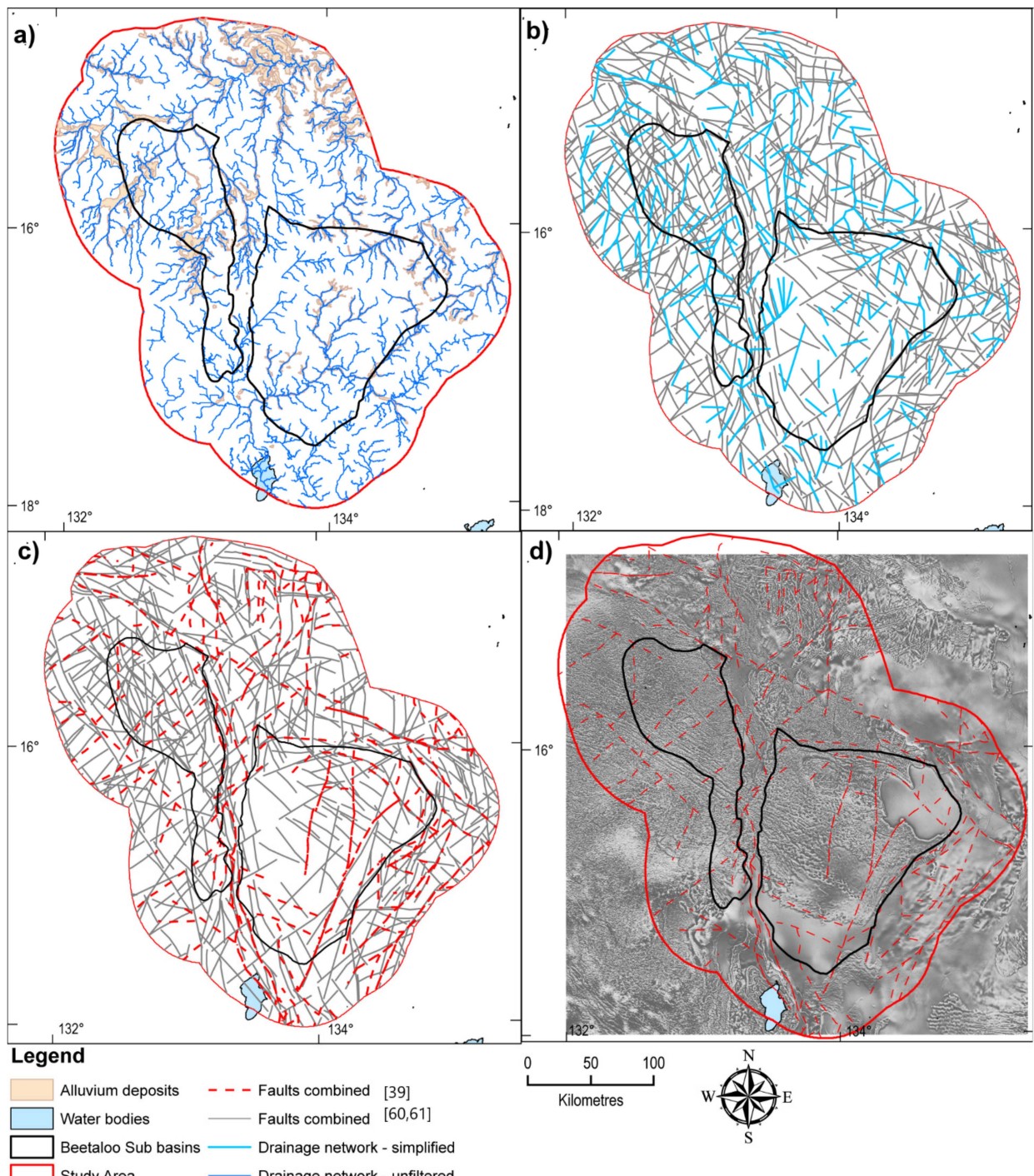

**Figure 2.** Unfiltered surface drainage network (**a**) and simplified interpretation (**b**) within the context of fault lineament datasets (**c**) and magnetic image of the first vertical derivative ((**d**)-details in Section 4.1) where lineaments were identified Simplified drainage network adopted approach: small segments with less than 10 nodes were automatically removed in SKUA (Paradigm/Emerson Trademark) and one iteration of smoothing was applied that further removed small irregularities without changing the overall orientation of major drainage lines. This output was then overlayed upon satellite and magnetic images where lineaments are evident for checking and quality control. In this process, minor mismatches of the automated simplified lines were removed and small drainage segments were connected along major lineaments for further simplification and reduction of the dataset.

## 4. Results

### 4.1. Aeromagnetic Imaging of Potential Structural Pathways

Figure 3A is a reduced-to-pole map of the Beetaloo GBA region, low pass filtered with 10 km cut off. By firstly applying the reduction-to-pole transformation, the asymmetric effect of the global magnetic field is removed, generating a transformation in which the edges of the magnetic anomalies are located under causative magnetic sources. However, if there is a significant remanent magnetisation component with different magnetic field properties compared to the assumed global field values, the transformation will be unsuccessful. Low pass filtering with 10 km cut off removes all wavelengths shorter than 10 km. This value is not directly connected with the depth of the magnetic source, but it is more about the size of the causative magnetic source. This map can be interpreted as displaying the contribution of the magnetic sources larger than 10 km. We assume that the size of magnetic sources increases with depth which motivates this relation between size and depth. As the total magnetic signal is dominated by the volcanics of the Kalkarindji Suite, this low pass filter has partly removed the magnetic signal of the basalts. The filtered image is potentially showing the magnetic content, including faults, caused by the lithological variation below the volcanics.

The first vertical derivative of the reduced-to-pole data (Figure 3B) emphasises the high-frequency (short wavelength) features which are usually produced by shallow magnetic sources. In the Beetaloo Sub-basin, this transformation enhances textural variation in the volcanics of the Kalkarindji Suite. The most prominent features are magnetic lineaments which can be caused by faulting. In the eastern portion of the Beetaloo GBA region, we can see the limits of the volcanics of the Kalkarindji Suite where the signal changes from rough to smooth. Additionally, there are several circular low magnetisation anomalies, approximately two kilometres in diameter, aligned with northwest trending lineaments. Some of these circular features can be observed on the Daly Waters Arch and its immediate surroundings.

### 4.2. Seismic Imaging of Potential Structural Pathways

The quality of the seismic signal above the explored unconventional plays is highly variable over the Beetaloo GBA region; this is due to the different seismic acquisition technics and processing applied in each survey, as well as the fact that our interval of interest is shallow and not fitting with the interval of interest the companies tried to best image with those datasets. We mapped the seismic signal quality along the Base Cambrian seismic horizon to minimise the risk of overinterpreting a poor signal and to provide a graphical estimation of the uncertainty (Figure 4). The Base Cambrian seismic horizon is a regional unconformity mainly corresponding to the base of the volcanics that is the shallowest and the least faulted of the surfaces imaged by the seismic. It is a difficult horizon to interpret in places as it is shallow where the signal can be poor. The main zones of poor imaging of the Base Cambrian are located at the centre of each sub-basin that are the deepest parts.

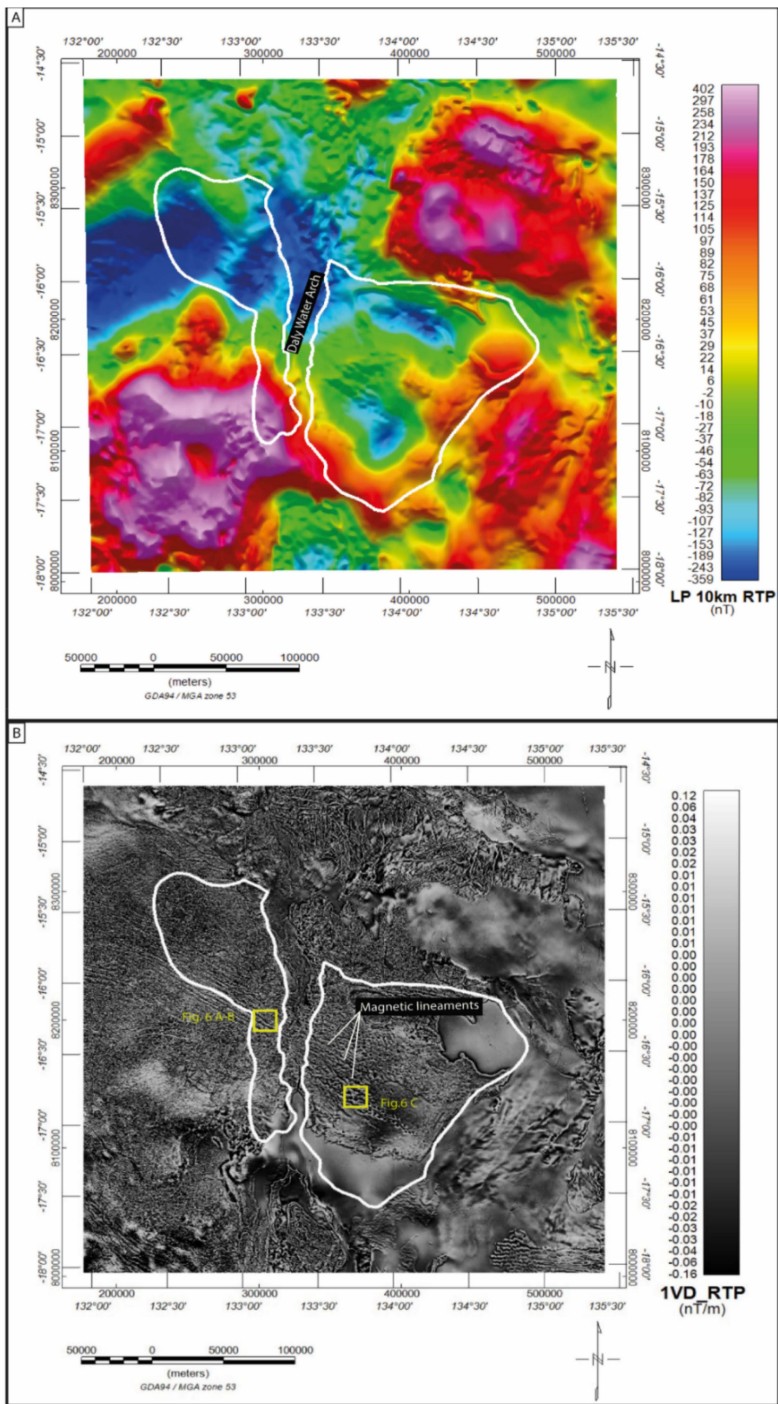

**Figure 3.** (**A**) Low pass reduced-to-pole map of the Beetaloo Sub-basin. The image is presented in a histogram equalised colour scale with shaded relief illuminated from the NE. The Sub-basin outline is in white. (**B**) The first vertical derivative of the reduced-to-pole data. The image is presented in a histogram equalised grey scale with shaded relief illuminated from the NE. The basin outline is in white.

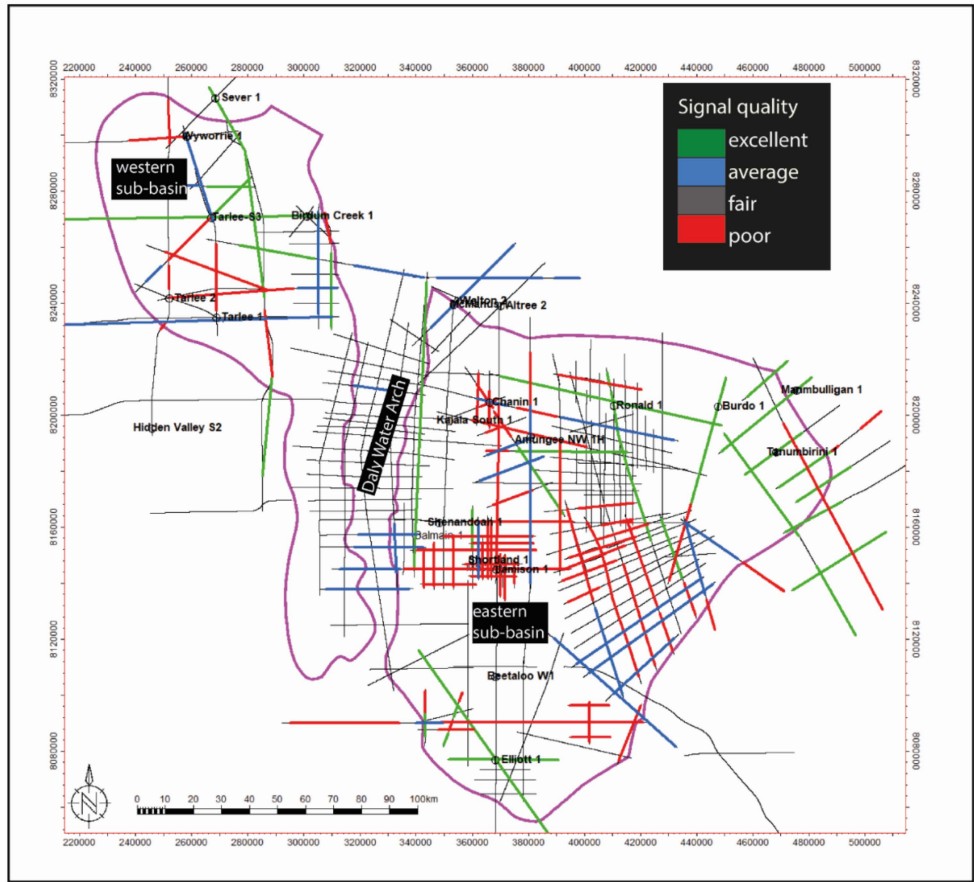

**Figure 4.** Interpretation of the 2D seismic reflection data quality over the Beetaloo Sub-basin.

Five horizons, from the Base Nathan to the Base Cambrian, and the associated fault networks, have been reinterpreted on the TWT seismic dataset and converted to depth with the parameters presented in Section 3 (Figure 5). Except for the northwest and north-northwest trending fault segments of the Daly Water Arch faults, the Birdum Creek fault and the OT-Downs fault (location of those major fault systems on Figure 1A) that have been active since the Paleoproterozoic and are still influencing the geological history to this current day, most faults that affect the basement up to the Moroak Sandstone, the formation above the unconventional plays, do not show signs of post-Cambrian unconformity reactivation on this seismic reflection dataset. The fault segments that have been active post-Cambrian are north-northwest trending (strike 110 to 160 deg) and are mainly located in the central part of the western Beetaloo Sub-basin and the northern and southern parts of the eastern Beetaloo Sub-basin. Those post-Cambrian faults that are rooted on the deeper fault system within the Beetaloo Sub-basin are subtle and not likely to have a strong strike slip component.

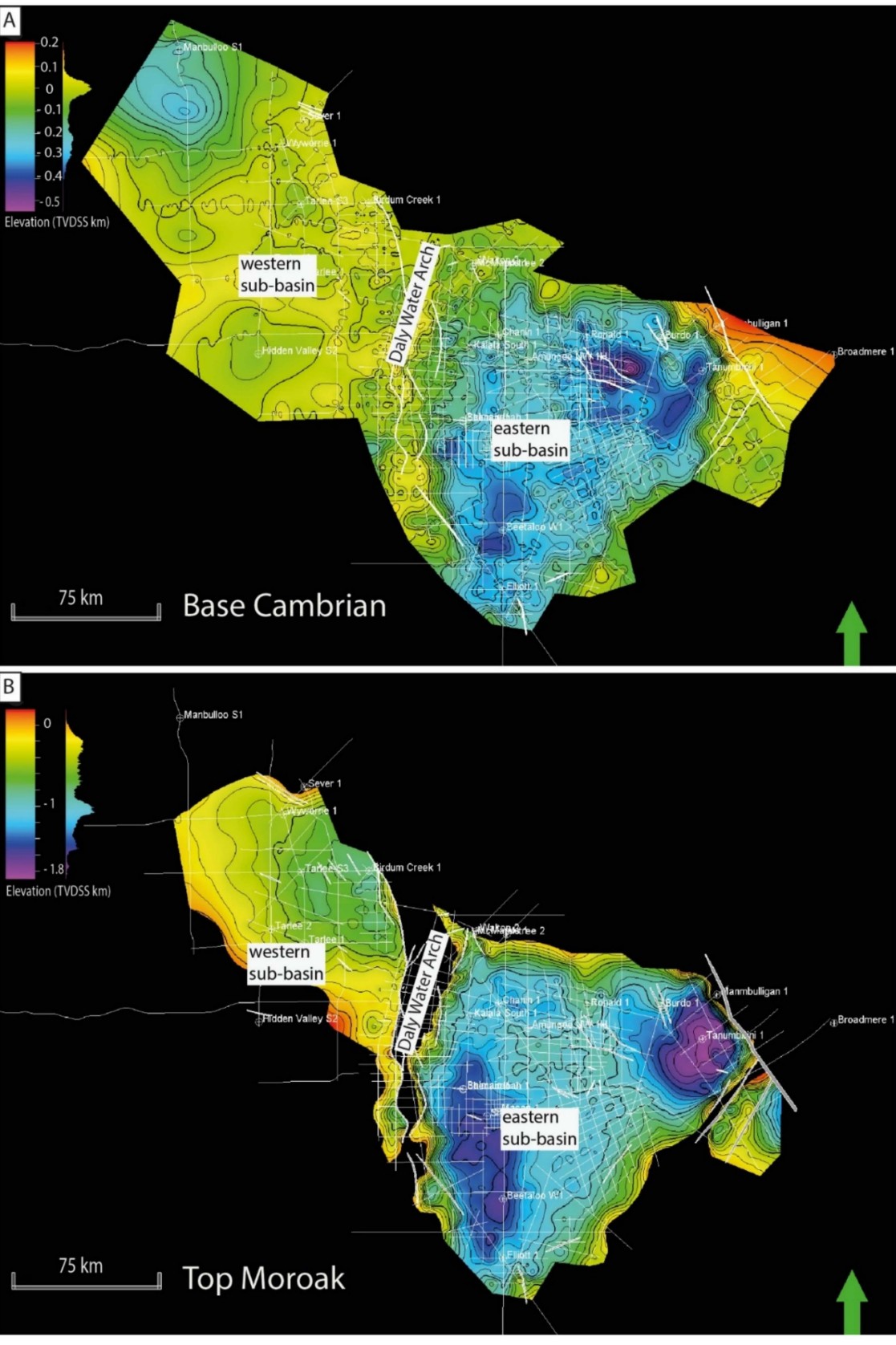

**Figure 5.** *Cont.*

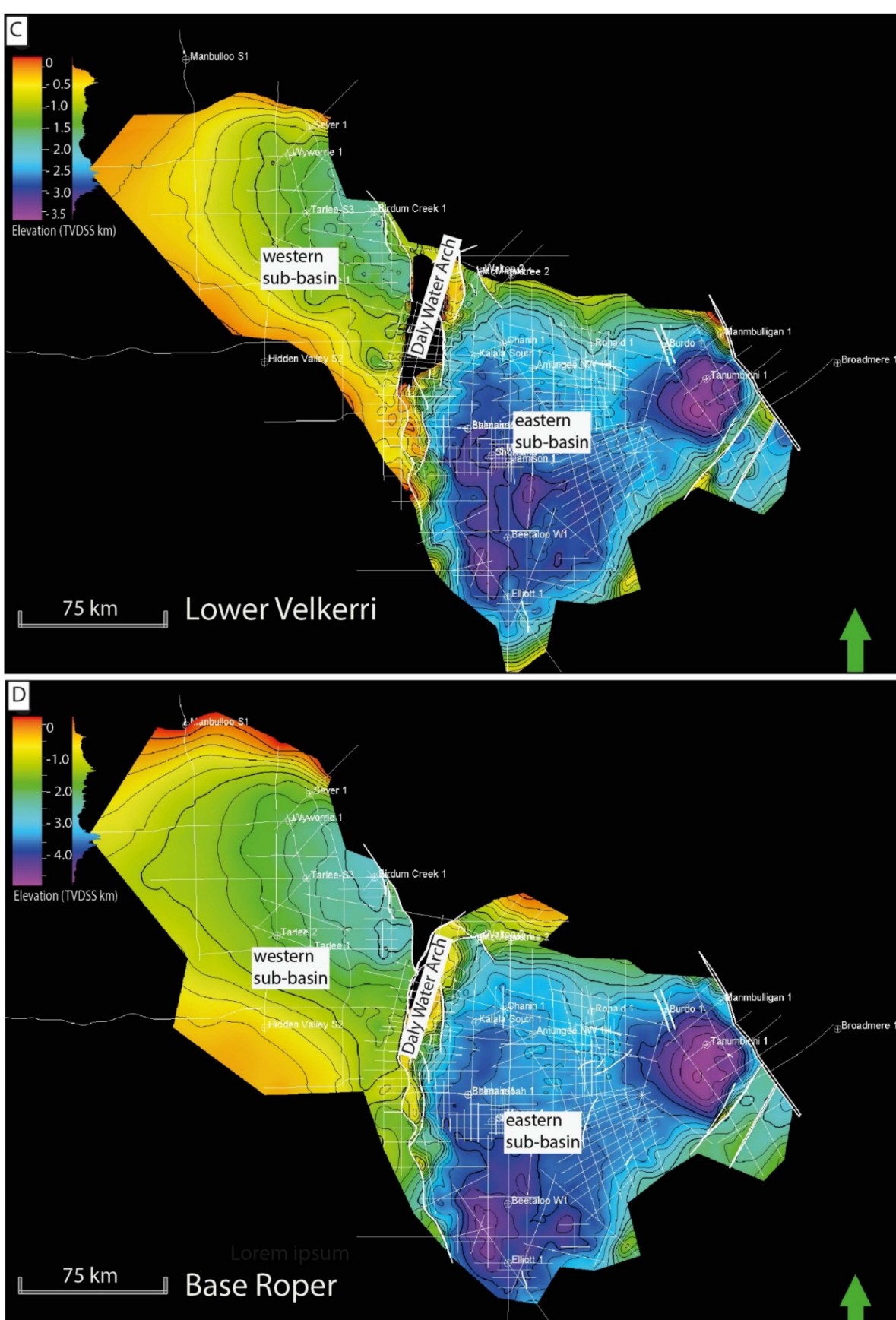

**Figure 5.** *Cont.*

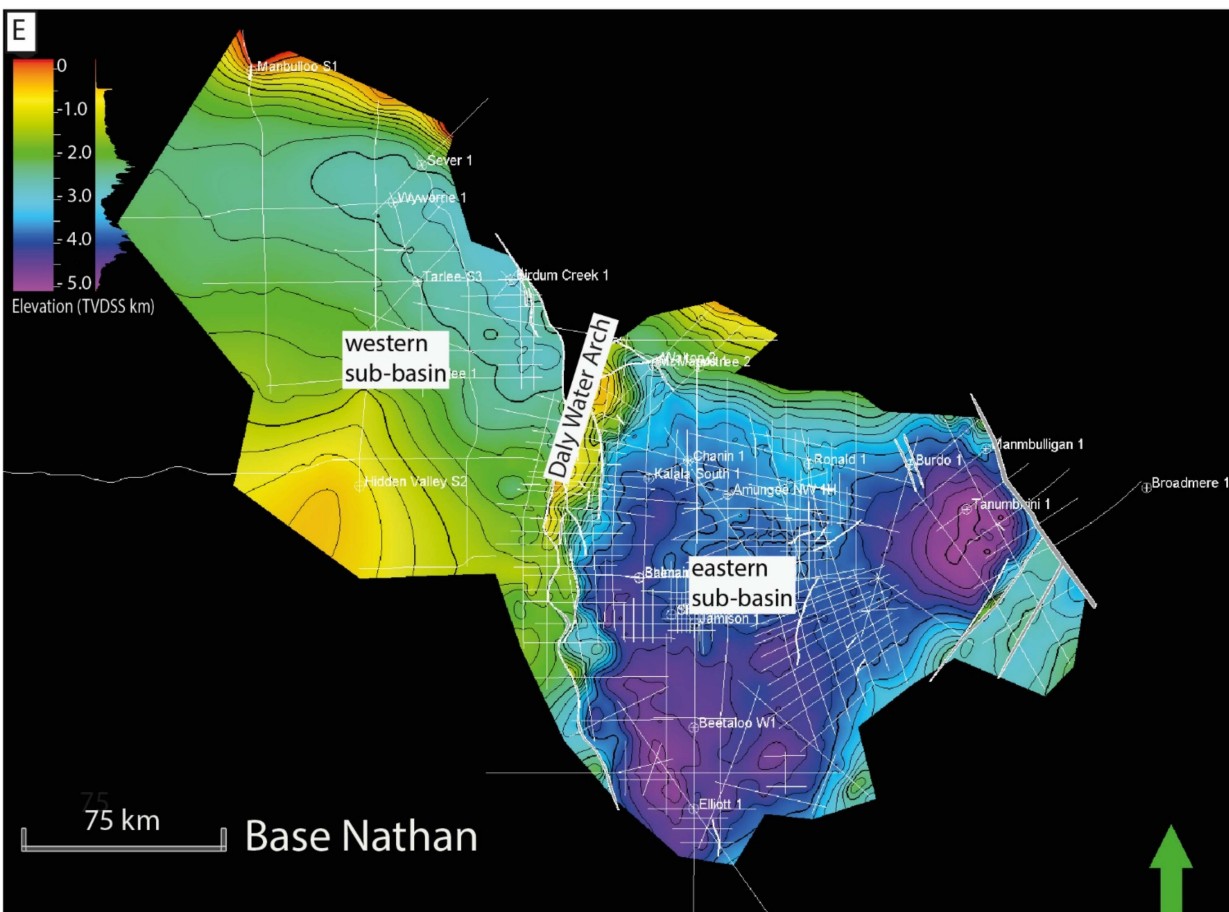

**Figure 5.** Depth-converted structural maps (TVDSS, m) interpreted in the Beetaloo Sub-basin for five horizons and fault network with a focus on the heritage faults reactivated post-Cambrian. From shallow to depth: (**A**) Base Cambrian; (**B**) Top Moroak Sandstone; (**C**) Lower Velkerri Formation; (**D**) Base Roper Group; (**E**) Base Nathan Group. The fault network is in bold white, the seismic lines in white.

We reviewed the seismic data in detail at the intersections with north-northwest trending lineaments observed on the magnetic dataset (Figure 3B). Some magnetic lineaments located in the central part of the western Beetaloo Sub-basin can be correlated along several seismic lines with seismic vertical discontinuities. Those seismic discontinuities (Figure 6A) have a low offset, visible as strong magnitude seismic reflector located between the Base Cambrian and the Moroak Sandstone as well as in the Base Cambrian reflector, and are associated with a deeper clear offset of the Derim Derim Dolerite. These vertical discontinuities are consequently rooted on top of a deep fault system and may be faults that have been slightly reactivated post-Cambrian. However, no indication of strike-slip movement is visible in those images (Figure 6). In this region, the seismic signal is of poor quality, discontinuous, and chaotic, and most of the lines located in the area of the north-northwest trending magnetic anomalies are not interpretable (Figure 6B).

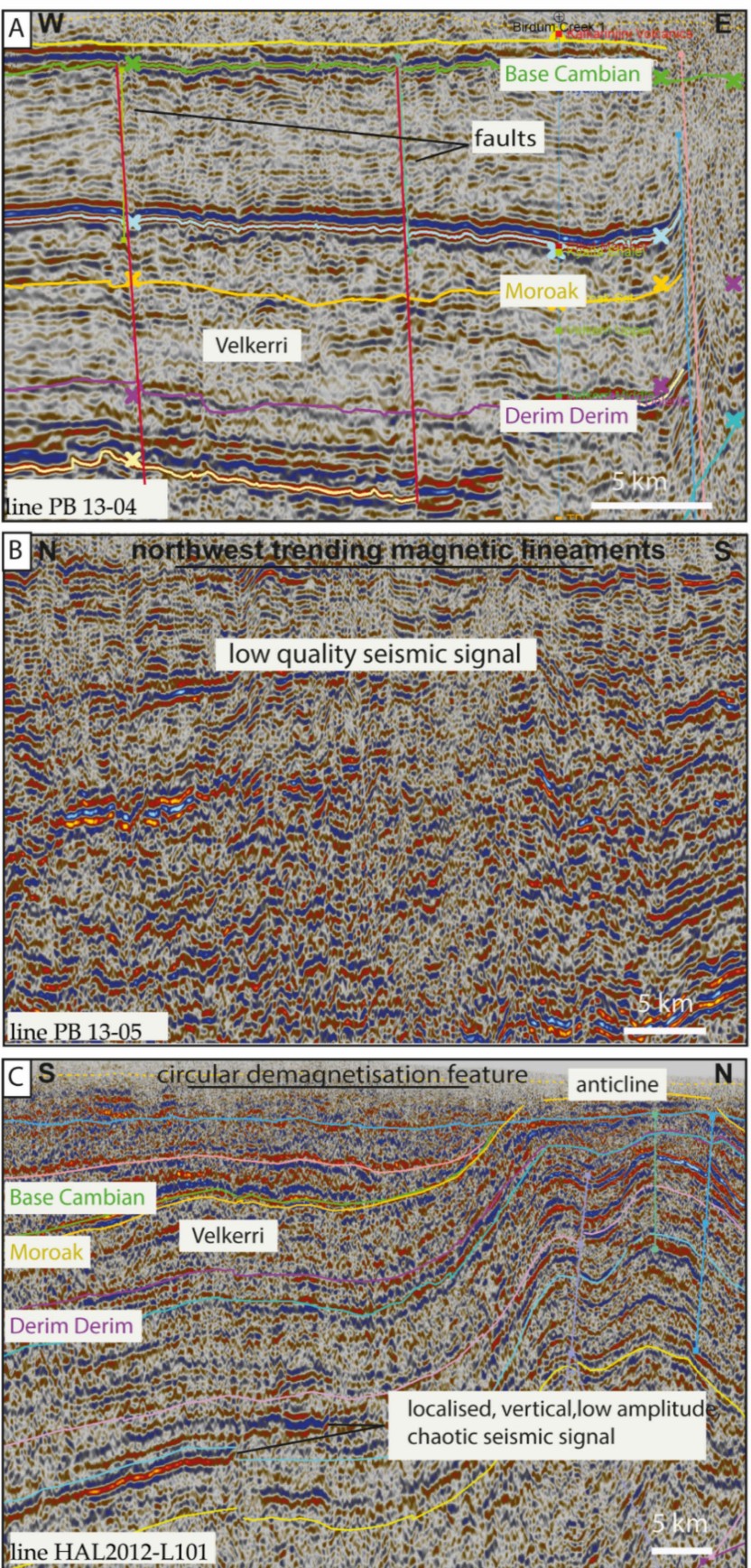

**Figure 6.** Details of the seismic interpretation in correlation with the magnetic interpolation at the top of the volcanics, see location on Figure 3B. (**A**) line PB 13-04; (**B**) line PB 13-05; (**C**) line HAL2012-L101.

The interpretation of the seismic data in the south-central part of the eastern Beetaloo Sub-basin highlights that the circular demagnetisation features observed at the top of the volcanics are located on the edge of an anticlinal feature (Figure 6C), in a zone were the Base Cambrian and above formations pinch out onto the Moroak Sandstone. In this area, the Velkerri Formation is located directly below the post-Cambrian sediments.

Zones of discontinuous, low magnitude, and localised seismic signals have been identified outside of the poor-quality seismic area previously identified (Figure 4) and can be interpreted as possible fluid or gas escape features/pipes or migration artefacts (Figure 7) as the presence of fluid or gas can locally alter the seismic signal. Most of those signals are located in the northern part of the western sub-basin and along the north-east trending fault boundary (Hot Spring Valley area) and in the south (Elliot area) of the eastern Beetaloo Sub-basin. We selected only localised discontinuous seismic signals that are repeated along seismic lines and are associated with faults or old wells. In the eastern sub-basin, the potential fluid or gas escape pathway features are aligned along the most recent faults: post-Wilton west-northwest trending reverse faults to the post-Wilton north-northwest trending strike-slip faults reactivated post-Cambrian unconformity.

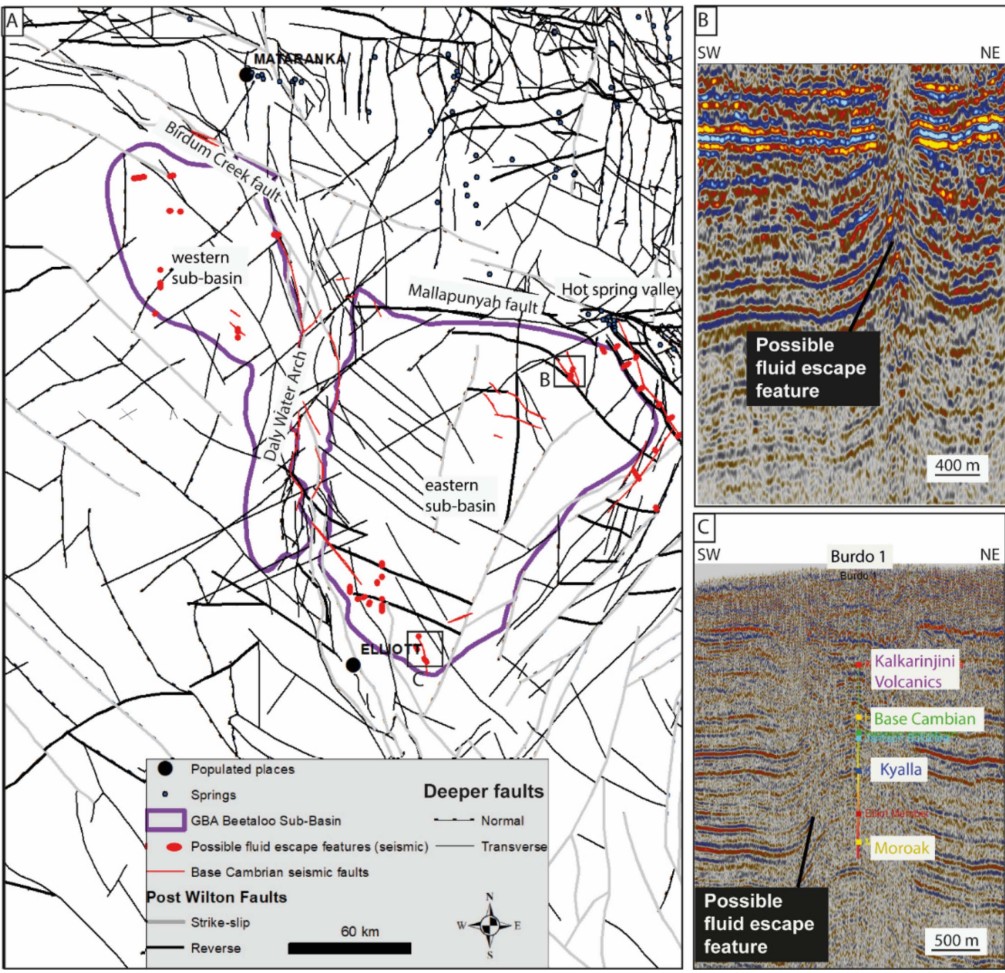

**Figure 7.** Potential fluid or gas escape features localised on the seismic reflection profiles of the Beetaloo Sub-basin. (**A**) location of the possible fluid escape features on a map view with the interpreted traces of the deep faults reactivated post-Cambrian, the post-Wilton faults [48], and the location of the surface springs. (**B**,**C**). Examples of potential fluid escape features on seismic reflection lines respectively in the northeast and southern parts of the eastern Beetaloo Sub-basin (lines MD92-58 and 89-20).

### 4.3. Helium Concentrations and Isotopic Composition Measured in Groundwater

Helium concentrations measured in groundwater and springs (dataset available as Supplementary Materials; Figure 8) ranged from solubility equilibrium ($4 \cdot 10^{-8} \cdot cc(STP)/g$) to three orders of magnitude above this value (maximum measured value $2.8 \cdot 10^{-5} \cdot cc(STP)/g$ in a well screened in the Antrim Plateau Volcanics). Results of regional multi-tracer studies [29,49,50] suggested that such high helium concentrations are not produced in situ in the Cambrian Limestone Aquifer, since uranium and thorium concentrations in host rocks of this aquifer are too low and residence times in groundwater are too short. The helium isotopic composition ($^3He/^4He$ ratio) indicated no helium source from the deeper mantle but only radiogenic origin for the elevated helium. The helium isotopic composition ($^3He/^4He$ ratio) together with the neon and helium concentrations indicated no helium source from the deeper mantle but only radiogenic origin for the elevated helium. This is because on a plot of $^3He/^4He$ versus Ne/He (see Supplementary Materials) the data fall on a clear mixing line between atmospheric helium (Ne/He of 3.9, slightly depending on temperature, $^3He/^4He$ of $1.36 \times 10^{-7}$,) and pure crustal helium (Ne/He ratio of 0 (*y*-axis intercept) and $^3He/^4He$ of $2 \times 10^{-8}$). This is true without any correction of the data (e.g., for excess air) within their given 1-sigma uncertainty. This indicates that the observed elevated helium concentrations originate from the sedimentary rock in the Beetaloo region rather than from fractures in the basement. However, the elevated helium concentrations show no systematic pattern of increase with flow distance (as expected for in situ production along the flow path) but a rather patchy regional pattern (Supplementary Materials; Figure 8). The following discussion elucidates how much the measured helium concentrations can contribute as another line of evidence to the geophysical interpretation of fractures being supportive for presence or absence of deep fluid movement.

### 4.4. Trend Analysis of Geological Structures and Surface Water Drainage

The comparison of spatial trends in the surface water drainage network and the fault lineages provided further indications on recent tectonic activity (Figure 9). We define as recent tectonic activity the events that are recorded at the surface and can alter the surface with active tectonic features, such as modifications of the drainage system. Two dominant azimuths (N-S and NNW-SSE) can be noted from the structural lineaments presented by [39,60]. These directions coincide with one of the major trends in surface drainage lines. This indicates that part of the drainage network within the Beetaloo region is actively influenced by the underlying geological structures, as also observed in other geological basins in Australia, such as the Cooper Basin [59].

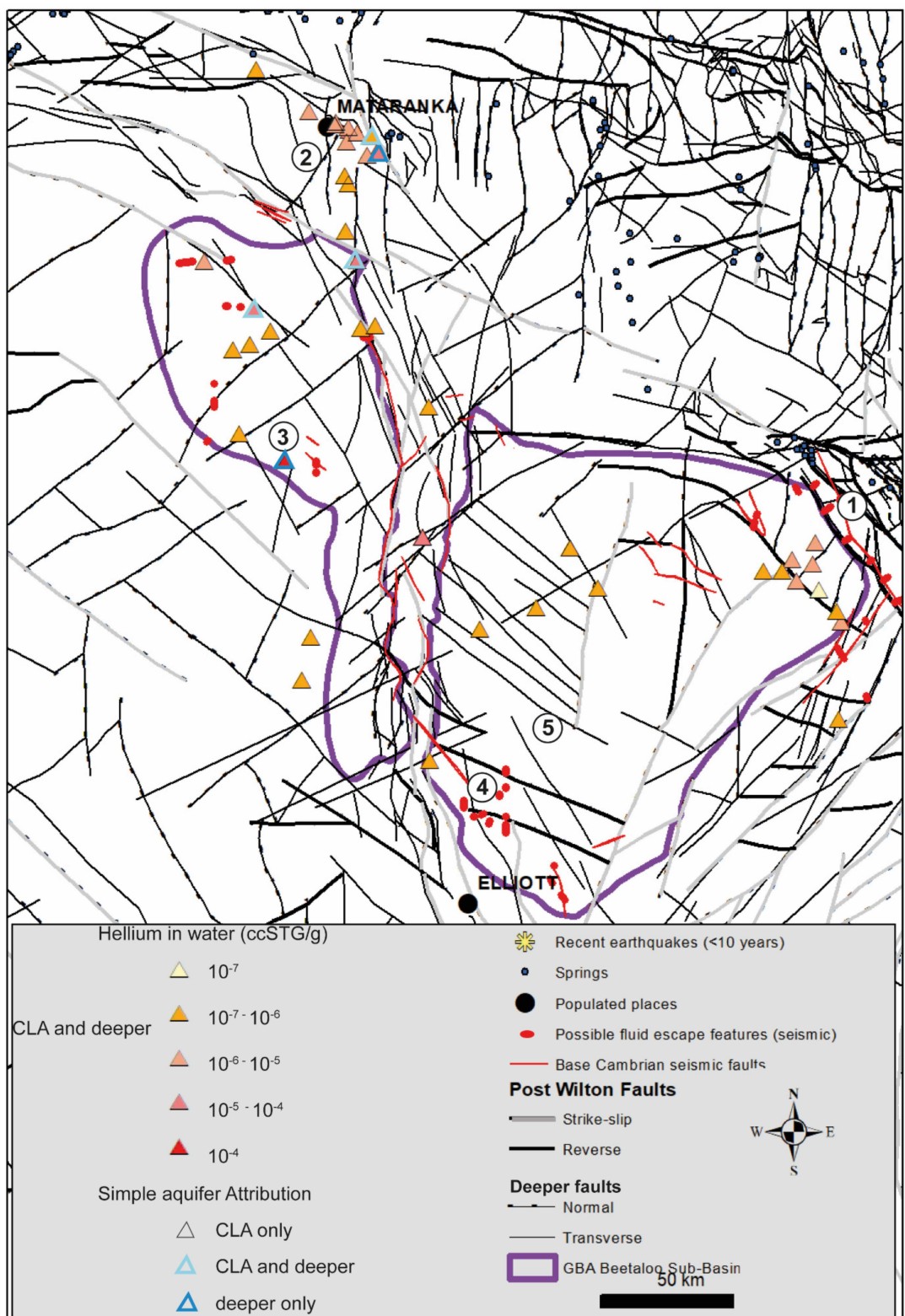

**Figure 8.** Potential current and post-Cambrian paleo fluid flow (1 to 5) associated with fault pathways. Compilation of this article datasets. 1 OT Downs fault zone, 2. Mataranka and Birdum Creek Fault, 3. Western Beetaloo Sub-basin, 4. southern part of the eastern Beetaloo Sub-basin, 5. Central part of the eastern Beetaloo Sub-basin.

Structural lineaments from the magnetic data (n=213)

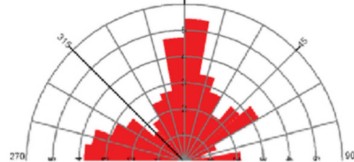

Structural lineaments from the SEEBASE (n=681)

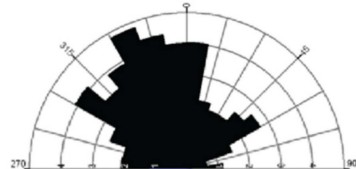

Surface drainage network from this study (n=130)

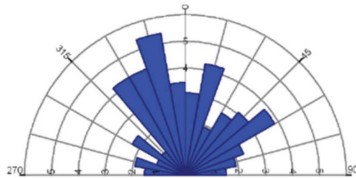

**Figure 9.** Published structural lineament from the magnetic data and SEEBASE ([39,60] analysis and this study unfiltered surface drainage network (see Figure 2)).

## 5. Discussion

### 5.1. Episodicity of Opening and Closure of Structural Pathways for Fluid and Gas through the Beetaloo Sub-Basin Tectonic History

The Beetaloo region underwent a complex tectonic history for more than 1000 million years, with alternating extensional and compressional stress regimes [27]. Few episodes of magmatic intrusions and flows from the depth to the surface or migration from one reservoir to another have been directly documented or are likely to be associated with certain types of deformation events and structural features.

Unconventional petroleum geological units, such as the Velkerri Formation, are part of the 5000 m thick Roper Group and were deposited during the Mesoproterozoic, 1440 Mya, post Syn-Isan Orogeny during an extended period of cooling [38] that may be associated with the separation of the North Australian Craton from Laurentia [62]. The extensional tectonic setting favoured magmatic intrusion events, such as the Derim Derim Dolerite emplacement, between 1300 and 1200 Mya [63] that could be associated with the final break-up of the Nuna (Columbia) supercontinent [64,65] and associated with episodic pulses of magmatism [66].

The middle to late Mesoproterozoic is a period of non-deposition that can be linked with uplift after plume-related magmatism [67] and a regional inversion known as the 'Post-Roper Inversion' ([68]. This inversion is not well constrained in the Beetaloo Sub-basin and might be related to any of the Mesoproterozoic tectonic events affecting central Australia in this period: the 1200 to 1150 Mya thermic and extensional Musgrave Orogeny and the 1090 to 1030 Mya extensional and hugely magmatic Giles Event [69–72]. The study of Precambrian petroleum inclusions entrapped in the Roper Group shows extensive hydrocarbon migration occurred during the Mesoproterozoic [73].

The dating of diagenetic illite, which forms in shales over the same temperature range as hydrocarbons, reveals the age at which the thermal maximum was reached in the Velkerri Formation [74]. These ages were $1017 \pm 23$ Mya and $980 \pm 23$ Mya in the eastern Beetaloo Sub-basin in Shenandoah 1A and Tanumbirini 1, respectively, and $1041 \pm 24$ Mya in the western Beetaloo Sub-basin in Tarlee S3 and correspond to the late stage of the Rodinia supercontinent amalgamation. The Beetaloo Sub-basin sedimentation was influenced by

the Rodinia Break up (860-570 My) that was regionally expressed by an east trending rifting and the deposition of the Centralian A Superbasin [75]. The 630-520 MyNorth-South Petermann Orogeny eroded those sedimentary super-sequences.

Proterozoic and Palaeozoic maximum burial scenarios were tested [76] for the Beetaloo Sub-basin and the results show that deep burial in the Paleozoic was of greater magnitude than burial on any of the earlier Proterozoic unconformities, suggesting that more than two kilometres of Proterozoic and Palaeozoic section, plus up to another 800 m of Mesozoic section, was eroded during the basin uplifts, providing changes in pressures and directly leading to new fluid circulation pathways through fault reactivation or opening.

The present study provides detailed information on potential episodes of fluid flow recorded in the last 500 My, post-Cambrian (Figure 8). Thermochronological analyses suggest that sedimentation in the Beetaloo region continued during the Paleozoic era [76–78] indicating a continuity of the extensional stress during that era before the modern compressional regime.

### 5.1.1. OT Downs Fault Zone

The major fault systems of the OT Downs fault zone (located in Figure 1), that is bounding the northern part of the eastern Beetaloo Sub-basin, is deeply rooted below the Nathan Group (Figure 5E). These regional sub-basin bounding fault systems are likely to have accommodated episodes of fluid flow as old as 1645-1640 My and are targeted for mineral exploration as potential fluid pathways for ascending metalliferous brines [79]. They are likely to have been reactivated through the more recent tectonic history of the region with current hot springs (e.g., Beauty Creek springs with a temperature of approximately 60 °C and Lagoon Creek springs) within the Hot spring valley located along the northern part of the fault zone (Figure 1). The presence of hot springs here confirms that a connection between the deep subsurface and the surface exists, and [40] suggested that a spring temperature of 60 °C requires active circulation to at least 650 m depth below ground surface along conduits with relatively good permeability. Furthermore, at the location of Beauty Creek springs within the Cox River catchment, a 90 degree stream diversion of Lagoon Creek also indicates the presence of faults and their control on the stream network development.

### 5.1.2. Mataranka and Birdum Creek Fault

In the Mataranka area, located a few tens of kilometres north of the Roper Group unconventional formation depocenter of the Western Beetaloo Sub-basin lobe, the measurements in the springs demonstrated elevated helium concentrations attributed to an upward flowing deep groundwater component (Figure 8; Supplementary Materials). Such a deep groundwater component needs a regional or local connection of the Cambrian Limestone Aquifer with deeper aquifers. However, the actual flux volume or source is unknown.

The springs are well aligned along a north-northwest trend (Figure 7), in a potential post-Wilton relay zone that: (1) fits with the post-Cambrian ENE-WSW fault trend interpreted in the Beetaloo region; and (2) is aligned with two recent magnitude 2 earthquakes recorded along a north-northwest trending strike-slip fault segment located in the north-western prolongation of the Birdum Creek fault in the Daly Basin, about 10 km away toward the south-west (Geoscience Australia, 2021) (Figure 8). The seismic interpretation clearly demonstrates a seismic activity of the north and northwest trending faults since the Paleozoic, especially towards the west of the Daly Waters Arch in the McArthur Basin (Figure 5) [48]. This structural background and the low magnitude events recorded along similar fault trends could support slow motion seismicity, such as creeping or pressure-dissolution [3] that is comparable to the one observed at the Green River (Utah) cold bubbling springs, with a mix of deep and meteoritic water and gas [79,80].

### 5.1.3. Western Beetaloo Sub-Basin

This part of the region is poorly imaged with few seismic lines (Figures 4 and 5). On the seismic data, few short north-northwest trending fault segments are aligned with continuous lineaments observed on the magnetic images at the top of the volcanics of the Kalkarindji Suite, tending to confirm a tectonic origin; however, these seismic faults are subtle and show no strike slip component (Figure 6). The orientation of the major watercourses in this area are closely aligned with north-northwest trending structural lineaments [40]. Potential seismic leakage features as well as helium concentrations in water up to 10-5 cc (STP)/g are observed along north-west trending lineaments and may indicate a connection of the shallow aquifers with the Paleozoic aquifers (Figure 8, Supplementary Materials). However, high values of helium in water should be interpreted with caution as no indication of mantle helium has been found in the Beetaloo Sub-basin; moreover, the baseline of He content in the different deeper aquifers below the CLA is yet to be investigated [81].

### 5.1.4. Southern Part of the Eastern Beetaloo Sub-Basin

This region is also poorly covered and characterised by a lack of magnetic resolution (Figure 3). Post-Cambrian fault activity and potential fluid paleo leakage features have been observed on the seismic dataset (Figures 4 and 7). The fault segments that have been reactivated in the last 500 My are not all aligned with the northwest to north-northwest trending strike slip faults; however, only this type of segment correlates with potential recent leakage.

### 5.1.5. Central Part of the Eastern Beetaloo Sub-Basin

Circular demagnetisation features imaged in the central part of the eastern Beetaloo Sub-basin (Figure 3B) have been previously interpreted as eroded paleo-dunes or holes indicating the local degradation of the volcanics sheet of the Kalkarindji Suite at fracture intersections, which have acted as conduits for fluid flow [82]. Other interpretations of those circular demagnetisations can also be put forward, such as demagnetisations linked with kimberlite magnetic pipes [83,84]. Those features are in the southern continuity of NNW trending magnetic lineaments that correlate with seismic faults (Figure 6A) and linked with potential current seismic evidence of fluid or gas leakage and high helium water content in the southern part of the western Beetaloo Sub-basin (Figure 8, Supplementary Materials). The seismic interpretation also showed that those potential paleo-fluid pipes are located along the flank of a NNW/SSE large anticlinal that cross-cut the eastern Beetaloo Sub-basin lobe and southwards of the Moroak Sandstone erosion (Figure 5).

The origin of the fluids related to the potential post-Cambrian paleo-leakages and current seepages observed at the surface are not clear. As previously mentioned, the post-Cambrian magnetic circular demagnetisation features are located on the southern flank of an anticlinal feature. In this region, in the southern part of the eastern Beetaloo Sub-basin, local extension could cause upward fluid circulation and the unconventional petroleum plays are located straight below the Cambrian unconformity, conditions favouring a connection between the deep Paleozoic aquifers and the unconfined aquifer or the surface.

### 5.2. Significance of Possible Circulation Episodes in the Context of Unconventional Petroleum Exploration

Episodic fluid and gas flows are currently observed in sedimentary basins spanning geological times. Several episodes of paleo-circulations of hydrocarbon, gas, or thermogenic fluids in reservoirs around inverted structures, uplifts, faults, fractures, and joints are attested world-wide by numerous evidence, such as geological formation bleaching [85,86], dissolution and re-crystallisation of iron-rich minerals [87,88], or other types of mineralized veins [89–91]. Studies of calcium carbonate en-echelon veins and travertine veins in fault zones show that circulation events are triggered by tectonic events and, once the fault is open, the duration of the circulation is dependent on external parameters that can be climatic, hydrological, or tectonic, such as the groundwater and meteoric water availability

and the pressure in the reservoirs [4,92–95]. The Mataranka and north Birdum Creek fault area (area 2, Figures 8 and 10), as well as OT Downs (area 1, Figures 8 and 10), are located outside of the Beetaloo Sub-basin, in areas strongly controlled by the basal north-west to north-northwest trending faults that have been episodically reactivated through the geological history of the McArthur Basin and are currently showing a strike slip component (Figure 5). Fault segments linked to those systems, especially around fault termination or relay zones, may be triggered and currently open acting as a conduit for deep fluids and gas leakage directly to the surface. As observed in the case of cold endogenic springs [96], this deep component can only represent a minor part of the water at the surface, as other shallow reservoirs, such as the unconfined CLA aquifer, also use the same conduit. However, the hypothesis of a complex system involving deep fluid or gas migration in regional formations cannot be excluded, since measured helium in groundwater indicated the presence of fluids from deeper formations in both springs and groundwater wells. Since neither the residence time of this deeper fluid nor helium concentrations in the deeper aquifers are available, this component cannot be quantified.

The Elliot area, in the southern part of the eastern Beetaloo Sub-basin (area 3, Figures 9 and 10) and the central part of the western Beetaloo Sub-basin (area 4, Figures 8 and 10), are located in zones that are prospective for dry gas plays in the Amungee member of the Velkerri Formation [12,26,27]. The post-Cambrian fault reactivation and paleo-leakages observed in those areas, as well as the potential recent leakage pathways observed on the seismic images and elevated helium content in groundwater (Figure 8), could indicate that those two zones have been reactivated and provide pathways for fluids and gas up to intermediate aquifers or the surface. However, the datasets in those areas are sparse and of low quality. Potential post-Cambrian paleo-leakage features (area 5, Figures 8 and 10) are observed on the magnetic data (Figure 3) in the central part of the eastern sub-basin and are located on top of the thicker unconventional depocenter that is prospective for dry gas in the Amungee member of the Velkerri Formation and in the Kyalla Formation [12,26,27]. Currently, the seismic interpretation and the helium measurements (Figure 8, Supplementary Materials) do not show any evidence of current reactivation of those pathways. The forthcoming Beetaloo Sub-basin seismic monitoring project [97], the Strategic Regional Environmental Baseline Assessment (SREBA) started in 2020 by the NT Government to understand the risks hydraulic stimulation poses to the NT [98] and the research on the social, economic, and environmental impact of NT's onshore gas industry lead by the CSIRO's Gas Industry Social and Environmental Research Alliance (GISERA) [99] should address those gaps.

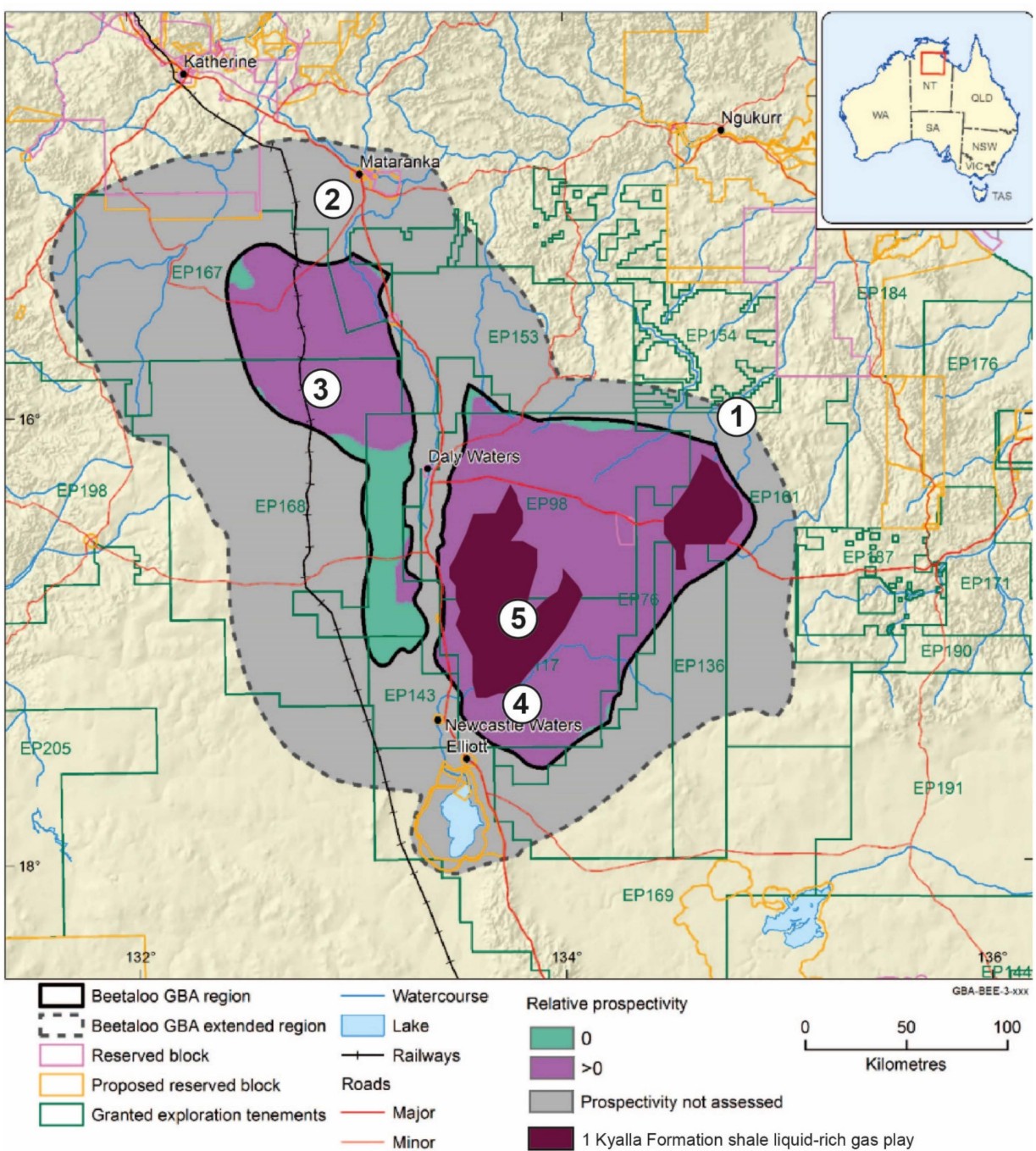

**Figure 10.** Unconventional petroleum prospectivity (27; 12) and potential current and post-Cambrian paleo fluid flow (1 to 5) associated with fault pathways. Compilation of datasets from this article. 1 OT Downs fault zone, 2. Mataranka and Birdum Creek Fault, 3. Western Beetaloo Sub-basin, 4. southern part of the eastern Beetaloo Sub-basin, 5. Central part of the eastern Beetaloo Sub-basin.

## 6. Conclusions

North to north-northwest trending strike slip faults have been reactivated in the recent geological history and are controlling the deposition at the edges of the Beetaloo Sub-basin. The Hot Spring Valley with active hot bubbling springs (approximately 60 °C) and the Mataranka springs (approximately 30 °C), located at the northern edge of the eastern Beetaloo Sub-basin and 10 km to the north of the western sub-basin, respectively, are linked to this system that is likely to locally connect the shallow unconfined aquifer with a deeper fluid source component. The presence of faults here is also suggested by significant stream

diversions (with a 90-degree deflection in stream course) in Hot Spring Valley, which is typically an indicator for active tectonics, and the correlation of stream course orientation with mapped faults.

The origin and flux of this deeper source is unknown and needs to be further investigated to assess if the input of helium measured in groundwater is local, with direct connection of the unconfined aquifer to deeper sources through the faults, or if regional circulation through stratigraphic connections involving a circulation within the Beetaloo Sub-basin could also be involved.

Few NW trending post-Cambrian fault segments are interpreted at the boundaries of the Beetaloo Sub-basin, in the prospective zones of dry gas plays in the Amungee member of the Velkerri Formation. The segments located in the northern part of the eastern Beetaloo Sub-basin show no evidence of modern leakages. The segments located around Elliot, in the south of the eastern Beetaloo Sub-basin, as well as low-quality potential faults in the central part of the western sub-basin, may have been recently reactivated and could act as open pathways of fluid and gas leakage from unknown origin. An origin from the mantle can be excluded on the basis of the measured helium isotopic composition and the Ne/He ratios. In those areas, the data are sparse and of poor quality, and further field work on the baseline emissions and groundwater tracers is necessary to assess if such pathways are currently active.

**Supplementary Materials:** The following are available online at https://www.mdpi.com/article/10.3390/geosciences12010037/s1. Supplementary File S1: Seismic figures; Supplementary File S2: Helium spreadsheet.

**Author Contributions:** Conceptualization, E.F.; methodology, R.C.; validation, A.D.; investigation, T.E., C.B.; C.G. original draft preparation, writing—review and editing, E.F., T.E., C.G., J.M. (Jelena Markov), J.M. (Jorge Martinez), M.R., C.T., A.S., C.W.; supervision, C.H.-H. All authors have read and agreed to the published version of the manuscript.

**Funding:** This research received no external funding.

**Data Availability Statement:** Not applicable.

**Acknowledgments:** This work was produced under the Geological and Bioregional Assessments program funded by the Australian Government Department of Agriculture, Water and the Environment. The Department of Agriculture, Water and the Environment, Bureau of Meteorology, CSIRO and Geoscience Australia are collaborating to undertake geological and bioregional assessments. For more information, visit http://www.bioregionalassessments.gov.au (accessed on 18 October 2021).

**Conflicts of Interest:** The authors declare no conflict of interest.

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
