# Peer review of "Fault-Related Fluid Flow Implications for Unconventional Hydrocarbon Development, Beetaloo Sub-Basin (Northern Territory, Australia)"

_geosciences, doi:10.3390/geosciences12010037_

Round 1

Reviewer 1 Report

General comment.

The authors present an interesting case study integrating geophysical, geological and hydrochemical data to map potential pathways for water and gases across complex sedimentary formations in the Northern Territories, in Australia. The approach used by the authors is innovative and allows to elaborate several hypotheses regarding the occurrence and movement of both paleo and more recent fluids. Results appear to be sound and supported by the illustrations. Figures, however, need to be significantly improved for the final manuscript (see detailed comments below). Overall, the implications of this study are important not only for industrial purposes (reservoir exploration and operation) but also for our basic understanding of subsurface flow processes over geological timescales. This study should therefore gather significant interest and is worthy of a prompt publication.

Figures.

  • Figure 1: Are colors in Figure 1 A related to the hydrostratigraphy or depicting topography? If the latter applies, make sure to provide a scale.
  • Figure 2: The left spine (or border) does not appear on the pdf reviewed. This figure could be improved by adding shaded satellite image to help the reader compare the mapped drainage systems (which were automatically mapped) and fault traces/lineaments with the real features observed.
  • Figure 3: The text (lines 258-259) mentions magnetic lineaments as prominent features, which can be caused by faulting. Such features are hard to visualize. It would certainly help if they were highlighted directly on the figure.
  • Figure 4: This figure could go in the appendix. It is good to let the reader know that not all seismic surveys provided the same quality, but this is more a technical point.
  • Figure 6: Put the title of each subfigure above the plot. It doesn’t read well otherwise.
  • Figure 7: What is special about the locations highlighted by the markers (crosses on Figure 7B and square markers on Figure 7C). At this scale, the reader can’t see much detail.
  • Figure 9: Remove ‘s’ from <10 yearss. Also, increase the thickness of the fault lines to make them more obvious.

Text.

  • Lines 271-272: Why is the signal quality so variable, and how were the categories (excellent, average and poor) defined?
  • Lines 323-324: The authors should explain why low magnitude local seismic signals can be believed to represent permeable pathways promoting water or gas migration.
  • Lines 339-357/Section 4.3: The map shown on Figure 9 that provides the location where groundwater and springs were sampled could be referred to.
  • Line 360: Here and throughout the text, the word ‘recent’ is used to describe some tectonic activity. It would be good define what recent means in the context of this study. An indication is provided in the Discussion (Line 414), but it would go to state this value upfront in the text.
  • Discussion: One general comment, could the use of other tracers (radon, of stable isotopes of water) yield also insight on the origin of the fluids sampled in this study?

Author Response

Thank you for your detailed review. We tried to improve the figures at best- attached is a detailed reply to reviews.

Reviewer 2 Report

Review

Frery et al.: Fault related fluid flow implications for unconventional hydrocarbon development, Beetaloo Sub-basin (Northern Territory, Australia)

Summary of content: The study investigates potential connections between unconventional petroleum plays and water assets in the Beetaloo Sub-basin in the Northern Territory, Australia.

Magnetic data and 2D seismic reflection profiles were used to image structural features, and fluid/gas leakage pathways. Helium content of the aquifer systems was sampled and measured. Recent fault activity was identified using a combination of sub-surface imaging and comparison of fault orientations and surface drainage networks.

The study concludes that there the shallow aquifers are connected to deep-seated gas source along faults, some of which show evidence of recent movement. Helium isotope data do not provide an unequivocal answer to whether the gas originates from the unconventional plays or a deeper source.

Although the authors point out that more data is needed to identify the source of the gas, the observations made in this study confirm the presence of fault-controlled fluid flow pathways connecting the stratigraphic levels where the unconventional plays are located with shallower aquifers the Beetaloo basin.

Review summary

Scientific aims clearly stated.

The text is well organized and well written.

Methods are sound and clearly described,

There are some shortcomings in documentation and presentation of results, which weaken the link between observations and conclusion. Development of the petroleum resources in this region is controversial and faces opposition from Traditional Owners as well as climate scientists, and environmentalists. One of the key issues in the Beetaloo Sub-basin, the risk of groundwater contamination from fracking-based production of the unconventional plays. The present paper is thus part of an ongoing debate and the findings presented here are likely to influence discussions and decisions by stakeholders. This fact stresses the need for clarity when presenting data and results. Improving the design of the figures and adding links/references to primary data sources would go a long way towards addressing this.

Conclusions appear supported by the observations made in this study, but presentation of results needs to be improved.

Detailed comments

Aeromagnetic survey data acquired over the last 50+ years were employed for mapping structural and lithological features.

Although the aeromagnetic data is said to be employed for structural interpretation, details somewhat sketchy. Reference is made to “…prominent […] magnetic lineaments which can be caused by faulting”, and “…several circular low magnetization anomalies, approximately two kilometers in diameter, aligned with northwest trending lineaments”. It would be nice to include a map with these features and lineaments interpreted from the aeromagnetic datasets. This would also tie the observations based on this dataset closer to the seismic interpretation (see below).

Seismic reflection 2D lines (1989-2015). 8500 km. Interpretation of shallow horizons and faults using all publicly available geological and geophysical data of the Beetaloo region. Fault polygons defined for five horizons. Depth conversion using check-shot velocities from 26 wells.

See comments to Figure 4. The depth maps provided in Figure 5 should be supplemented by isochore maps for the rock volumes between pairs of mapped reflectors. See also comments to Figure 7.

Helium measurements from groundwater samples. With respect to the helium measurements, no detailed documentation of where these samples were collected is provided beyond referring to Geological and Bioregional Assessment Program (2021a) Fact sheet 12, and Geological and Bioregional Assessment Program (2021b) Regional tracer results from the Cambrian Limestone Aquifer. These are both summary documents. Please provide a reference to the primary documentation.

Fault and surface drainage network mapping (tectonic geomorphology). The drainage network was extracted automatically. It is not clear if the “simplified” drainage network shown in Figure 2 was also extracted automatically.

The data is summarized in Figure 8 (see also comments to Figure 8 below). It is not clear if the surface drainage network line dataset is based on the “unfiltered” or “simplified” data shown in Figure 2. Since n=130, one can assume the latter. It is however not made clear how this “simplified” network was generated, but there appear to be a substantial number of instances where the “simplification” provides apparent mismatches with the original data (see examples included in comments to Figure 2 below). The accuracy of the resulting plot can therefore be questioned. If the “simplified” network was generated automatically, I encourage the authors to do a manual QC of the results.

-Line 303:  “We reviewed the seismic data in detail at the intersections with north-northwest trending lineaments observed on the magnetic dataset (Figure 4).”  No such lineaments are shown in Figure 4.

-Line352: “However, the elevated helium concentrations show no systematic pattern of increase with flow distance (as expected for in-situ production along the flow path) but a rather patchy regional pattern (Figure 8).” This should refer to Figure 9.

-Line 582. Link for supplementary materials does not work.

Figures

Figure 1A.

-Add scale.

-Add colour scale for map (depth).

-Well- and fault-labels largely illegible. Consider removing labels not referred to in the text.

-Poor visual discretization of main surface fault traces and Post Wilton geophysical faults (maybe use different colour for the two?)

- Replace white labels with black lettering with no labels and white lettering.

Figure 1B

-Consider adding a small inset table or schematic to showing the Cambrian Limestone Aquifer stratigraphy in the Georgina, Daly and Wiso basin. Although this is described in the caption, it is not easy to grasp for someone not familiar with the stratigraphy in the area without spending time some time with paper and pencil.

Figure 2

-Poor quality figure (low resolution) in the version available to the reviewer.

- Very, very overloaded figure, screen-dump from a mapping programme?

-Consider splitting into several maps, e.g. one showing unfiltered + simplified drainage network, and one showing faults).

-How the authors arrive at the simplified drainage network is not very clear

Figure 3

-Very small/illegible typeface on map coordinates – consider simplifying scales and use larger typeface.

-Replace the white label with black typeface with plain white typeface.

Figure 4

-Poor quality figure (low resolution) in the version available to the reviewer.

- Replace white labels with black typeface with white typeface.

-Many well-name labels are hard difficult to read/illegible. Consider replacing well names on the map with numbers at the well position (white circles, black typeface) and add a table in the legend listing number and corresponding name of the well.

-Thin grey seismic lines are not explained in the legend or caption. Why are these not classified in terms of signal quality?

-The outline of the Eastern and Western Beetaloo Basin in Figure 3 and 4 should be kept identical in order to position Figure 6 A, B and C in relation to the seismic lines shown in Figure 4. Alternatively the position of Figure 6 A, B and C should be indicated in Figure 4 as well as Figure 3.

Figure 5. Consider adding isochore maps for intervals between interpreted reflectors. These often highlight tectonic accommodation space creation better than maps.

Figure 6 Add scale(s).

Figure 7. Very poorly designed map (screen-dump?). Reviewer copy is also poor resolution.

Clean up the legend:

  1. a) Remove underscores,
  2. b) Correct use of small caps, large caps in labels.
  3. c) Correct spelling (e.g. frameworkboundaries to framework boundaries; populated places to Populated places/(Settlements?); PossibleFluidEscapeFeatures to Possible fluid-escape features etc. ).
  4. d) Explain/improve labels (what do you mean by “ntspr_2M_gw”? “Coast_10million”? “frameworkboundaries”? “State_Borders_10million”. Amend labels or explain in caption.
  5. e) Several items in the legend appear to have the same or very similar signatures.

Clean up map:

  1. Not possible to differentiate post-Wilton fault types properly in the figure.
  2. Could not find “Coast_10million”, “frameworkboundaries” or “State_Borders_10million” on the map. If these items are present, please use a more contrasting signatures to make it more visible. If they are not present on the map, remove these items from the legend.
  3. Consider replacing the purple outline of the eastern and western sub-basin with light grey shading.
  4. Consider using the most prominent colour (red) for the possible fluid escape features as this is the key element of the map. Bright green might also bee an option.
  5. Not possible to differentiate “ntspr_2M_gw” (whatever that is) and “PossibleFluidEscapeStructures” (same colour).

The link between interpreted faults and fluid or gas escape features is a key feature of this figure. The map differentiates between “BaseCambrian seismic faults”, Post-Wilton “Strike-slip” and “Reverse” faults, and “FAULTS”. The categorization is not adequately explained (e.g. what differentiates “FAULT” from a “BaseCambrianSeismic fault”?).

Consider adding a simple conceptual sketch highlighting the stratigraphic position of the different faults (and possible fluid escape structures– this would also help visualizing which stratigraphic intervals the potential fluid escapes originate from and which stratigraphic intervals are potentially connected along potential fault-related fluid flow pathways.

Figure 8. Please provide a more informative caption to what these plots show. What is included in the “Drainage and structural lineament analysis” plot?

Figure 9. Poor resolution in the copy provided for the review. Consider using higher contrast colours or thicker lines to highlight different faults on the map.

I hope you will find these comments useful when revising the manuscript and look forward to seeing the paper published in Geoscience.

Author Response

(The authors gave the same response as above.)

Round 2

Reviewer 2 Report

Review v2

Frery et al.: Fault related fluid flow implications for unconventional hydrocarbon development, Beetaloo Sub-basin (Northern Territory, Australia)

The reviewer's comments have largely been addressed by the authors. However, there are still a couple of minor issues.

With respect to the helium measurements, the references to the primary documentation requested by the reviewer in comments to v.1 of the manuscript are not provided, nor do the authors provide any comment as to why they deem this unnecessary. Section 4.3 Helium concentrations and isotopic composition measured in groundwater, is framed around results from previous studies. Providing references to these original studies (including access to the actual data that allow objective verification) rather than referencing to summary papers would be more in line with scientific publishing practice. If the primary documentation is not accessible, the authors should state this clearly.

Figure 4. Seismic reflection data quality is ranked and colour-coded as “Excellent”, “Average” and “Poor”. Where does the fourth category, “Fair”, which also is a qualitative category, fit in, and why not include this category in the legend?

Figure 5. Isochron maps have been included in the supplementary material, the initial review suggested isoCHORE maps. Including these is maybe not critical, but as mentioned in the initial review, thickness maps sometimes highlight structural lineaments quite well. I don't know if this is the case here, but if they do, including isochore maps would enhance the readers' understanding of basin geometry and -infill.

Figure 7. Although the design of the figure is very much improved compared to the one submitted initially, it is still difficult differentiate between the Post Wilton “Strike-slip” and “Reverse” faults. Please use contrasting colours to differentiate them.

If I understand the map correctly, fluid escape features on the map are shown as red dots. Why does the legend show these as thick red lines?

Figure 8. Please keep the number of measurements included in each plot. The plot “Surface drainage network from this study” is identical to the original figure and should have n=130 measurements. According to the new caption shows the unfiltered surface drainage network. To my understanding this should correspond to Figure 2 a, which shows substantially more than 130 lineaments. How did you pick the ones you measured? Please clarify.

Author Response

Reply to review

The authors would like to thank you the reviewer 2 for this second round of reviews. All comment avec been considerate: figures 1,4,7,8, 9 have been modified, an helium supplementary file have been added and the seismic supplementary file has been updated with the addition of the isochores. The replies to detailed comments are in purples, directly below each comment

Frery et al.: Fault related fluid flow implications for unconventional hydrocarbon development, Beetaloo Sub-basin (Northern Territory, Australia)

The reviewer's comments have largely been addressed by the authors. However, there are still a couple of minor issues.

With respect to the helium measurements, the references to the primary documentation requested by the reviewer in comments to v.1 of the manuscript are not provided, nor do the authors provide any comment as to why they deem this unnecessary. Section 4.3 Helium concentrations and isotopic composition measured in groundwater, is framed around results from previous studies. Providing references to these original studies (including access to the actual data that allow objective verification) rather than referencing to summary papers would be more in line with scientific publishing practice. If the primary documentation is not accessible, the authors should state this clearly.

The helium measurements have been added as supplementary data and the author made a clear difference between this study Helium measurements and other tracer studies we are citing.

Figure 4. Seismic reflection data quality is ranked and colour-coded as “Excellent”, “Average” and “Poor”. Where does the fourth category, “Fair”, which also is a qualitative category, fit in, and why not include this category in the legend?

Done, thank you

Figure 5. Isochron maps have been included in the supplementary material, the initial review suggested isoCHORE maps. Including these is maybe not critical, but as mentioned in the initial review, thickness maps sometimes highlight structural lineaments quite well. I don't know if this is the case here, but if they do, including isochore maps would enhance the readers' understanding of basin geometry and -infill.

The isochores have been added in the seismic supplementary dataset

Figure 7. Although the design of the figure is very much improved compared to the one submitted initially, it is still difficult differentiate between the Post Wilton “Strike-slip” and “Reverse” faults. Please use contrasting colours to differentiate them.

If I understand the map correctly, fluid escape features on the map are shown as red dots. Why does the legend show these as thick red lines?

Post Wilton “Strike-slip” and “Reverse” faults have been redrawn in all figures (Figure 1, 7, 9) with a new colour code, the fluid escape features are now shown as red dots in the legend as well.

Figure 8. Please keep the number of measurements included in each plot. The plot “Surface drainage network from this study” is identical to the original figure and should have n=130 measurements. According to the new caption shows the unfiltered surface drainage network. To my understanding this should correspond to Figure 2 a, which shows substantially more than 130 lineaments. How did you pick the ones you measured? Please clarify.

Number of measurements added in each plot. Figure 2a correspond to the SEEBASE structural lineaments (n=681).